# Intelligent Lesion Selection: A Novel Method for Longitudinal Assessment of Breast Cancer Lung Metastases

**Melika Qahqaie**[1,2]                                     MELIKA.QAHQAIE@FAU.DE
**Veronika A. Zimmer**[2]               VERONIKA.ZIMMER@SIEMENS-HEALTHINEERS.COM
**Eduardo Castañeda**[1,2]                          EDUARDO.CASTANEDA@FAU.DE
**Katariina Peltonen**[3]                        KATARIINA.H.PELTONEN@HUS.FI
**Joonas Laaksolilja**[3]                             JOONAS.LAAKSOLILJA@HUS.FI
**Juho Lähteenmaa**[3]                              JUHO.LAHTEENMAA@HUS.FI
**Tobias Heimann**[2]                 TOBIAS.HEIMANN@SIEMENS-HEALTHINEERS.COM
**Andreas Maier**[1]                                  ANDREAS.MAIER@FAU.DE
**Dominik Neumann**[2]               DOMINIK.NEUMANN@SIEMENS-HEALTHINEERS.COM

[1] *Friedrich-Alexander-Universität Erlangen-Nürnberg, Pattern Recognition Lab, Erlangen, Germany*
[2] *Digital Technology and Innovation, Siemens Healthineers, Erlangen, Germany*
[3] *HUS Helsinki University Hospital, Comprehensive Cancer Center, Helsinki, Finland*

**Editors:** Accepted for publication at MIDL 2025

## Abstract

Breast cancer, the second most common cancer globally, often metastasizes to the lungs, requiring frequent computed tomography (CT) scans to monitor disease progression. Manual analysis by radiologists is time-consuming and prone to variability, underscoring the need for automated systems to enhance accuracy and efficiency. The goal of such systems is to optimize processes like RECIST score calculation for tumor response assessment. This study presents a pipeline for the automated temporal analysis of breast cancer lung metastases. Existing lung nodule detection and segmentation models were adapted for detecting and segmenting breast cancer metastases. Registration-based lesion tracking was incorporated, and a novel Temporal Lesion Pair Classifier was developed to identify significant lesions and estimate tumor load evolution by summing their diameters, following an adaptation of the RECIST guidelines. Evaluated on a unique dataset of breast cancer patients, each with multiple annotated CT scans at different disease stages, the proposed pipeline demonstrated a 42% reduction in median tumor size progression discrepancy for consecutive study pairs and improved tumor response classification accuracy by 22% at the patient level.

**Keywords:** Longitudinal disease assessment, Lung metastasis in breast cancer, Computed Tomography (CT), Deep learning, Lesion detection and selection

## 1. Introduction

Breast cancer affects over 2.3 million individuals annually and is a leading cause of cancer-related death among women worldwide (WHO, 2023). It commonly metastasizes to specific organs, with the lungs being the second most frequent site after bones (Yang et al., 2022; Wang et al., 2019). The prognosis for lung metastases is poor, with a 5-year survival rate of 16.8% (Schuler and Murdoch, 2021), necessitating continuous CT scans to monitor disease progression and evaluate treatment effectiveness (Yang et al., 2020).

In clinical trials, RECIST 1.1 guidelines are a framework widely used for measuring solid tumors and assessing changes in tumor size over time (Eisenhauer et al., 2009). Under RECIST , two to five target lesions are selected at baseline, with no more than two lesions per organ. These target lesions must be measurable, with non-nodal lesions requiring a minimum diameter of 10 mm as assessed by CT scans. The sum of diameters (SoD) of these target lesions is calculated at baseline and re-evaluated at each follow-up. Non-target lesions are monitored to determine stability, progression, or disappearance. The objective tumor response is categorized as complete response (CR), partial response (PR), stable disease (SD), or progressive disease (PD). PR indicates a $\geq 30\%$ reduction in SoD, PD an increase of $\geq 20\%$ or the appearance of new lesions, and SD reflects changes within these thresholds. CR is defined as the disappearance of all significant lesions. Automatic calculation of clinically relevant criteria, such as RECIST, requires robust methods for lesion detection, segmentation, tracking, and assessment across multiple time points.

Previous works on longitudinal analysis in cancer types and organs other than breast cancer have demonstrated the potential of automated methods. In (Mukherjee et al., 2024), the focus was on lesion matching across scans with varying annotations and scan parameters using image registration and the Hungarian algorithm, achieving accurate lesion correspondence. In (Venkadesh et al., 2023; Li et al., 2023), deep learning approaches have leveraged temporal information to improve malignancy predictions. Only few works concentrate on RECIST score estimation, which is essential for assessing treatment response in clinical settings. In (Zhou et al., 2024), a pipeline was developed for RECIST score estimation in liver cancer integrating lesion detection and image registration methods. This pipeline was trained on comprehensively annotated liver lesion data, with RECIST scores calculated by selecting target lesions from detected lesions. However, its performance was only evaluated on liver tumors, limiting its applicability to other cancer types.

Factors such as lesion size, morphology, and growth rate are critical for cancer prognosis, however, many studies do not consider temporal dynamics (MacMahon et al., 2017; Liao et al., 2019). Indeed, most current approaches for breast cancer metastasis focus on single time-point analyses, neglecting the need for longitudinal assessments critical for monitoring disease progression and assessing tumor burden over time (Moreau et al., 2021; Li et al., 2023). While the analysis of lung metastasis is crucial for evaluating disease progression in breast cancer, research in these areas remains limited compared to studies on bone and lymph node metastases (Yang et al., 2020; Liu et al., 2021; Moreau et al., 2020).

Methods developed for lung cancer are often not directly applicable to breast cancer metastases in the lung. Pulmonary metastases from breast cancer frequently present as numerous well-defined nodules, whereas primary lung cancers typically appear as solitary, irregularly shaped nodules (Stana et al., 2025). Existing datasets are often not designed to address the unique characteristics of breast cancer metastases in the lungs, posing additional challenges in developing robust, generalizable models.

This work presents an automated system for the temporal analysis of breast cancer metastases in the lungs using longitudinal 3D CT data. We leverage existing single-timepoint lung nodule detection and segmentation models trained on lung cancer images to detect and segment breast cancer metastases. Lesion tracking is performed using image registration. A novel Temporal Lesion Pair Classifier (TLPC) is introduced to identify temporally significant lesions for the automatic estimation of a RECIST-like score to assess

disease progression. The ultimate goal is to provide a reliable, efficient, and precise tool for clinical decision support in the management of metastatic breast cancer.

The key contributions of this study are two-fold. First, a complete pipeline for automated longitudinal analysis of metastatic lesions, integrating proven single-timepoint analysis modules, is proposed. Second, a novel Temporal Lesion Pair Classifier to identify significant lesions for estimating disease progression in alignment with an adaptation of the RECIST guidelines is proposed. The pipeline is evaluated on a unique dataset of breast cancer patients, with an average of 4 scans per patient. An expert radiologist identified and annotated up to 15 of the most significant lesions per patient, focusing on those showing notable growth or shrinkage.

## 2. Methods

### 2.1. Pipeline Description

The pipeline for temporal analysis of disease progression (shown in Figure 1) consists of five submodules: (i) lesion detection to identify potential candidates, (ii) lesion segmentation to determine lesion boundaries, (iii) lesion tracking, which aligns scans from consecutive studies and matches detected lesions to form lesion pairs for tracking changes over time, (iv) lesion pair identification using a novel method called the Temporal Lesion Pair Classifier (TLPC), which categorizes lesion pairs as either *Significant* or *Insignificant*, and (v) longitudinal analysis, where only pairs classified as *Significant* are considered, ensuring that the system focuses on clinically relevant lesions.

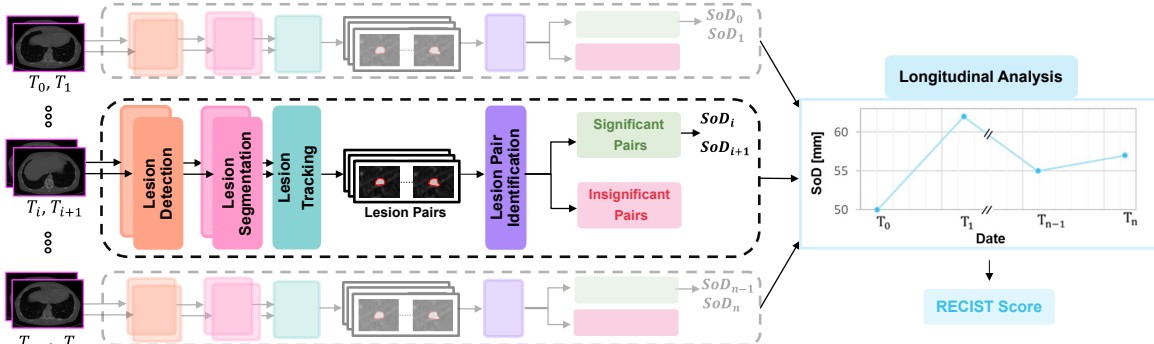

Figure 1: Overview of the proposed pipeline for automated longitudinal assessment of breast cancer lung metastases.

### 2.2. Lesion Detection and Segmentation

The lung lesion detection process leverages a proprietary system developed by Siemens Healthineers, originally designed for lung cancer detection. This system is an adaptation of the baseline two-stage nodule detection framework presented in (Liu et al., 2020), with modifications to enhance lung lesion detection. It utilizes a RetinaNet-based detector (Ross and Dollár, 2017) to identify candidate nodules, followed by candidate classification using

an ensemble of DenseNet3D (Huang et al., 2017) and EfficientNet3D (Tan and Le, 2019) models, which generates nodule classification scores, helping to identify detected candidates that are more likely to be nodules. The models were trained on a variety of datasets, including NLST (Team, 2011), LUNA (The Cancer Imaging Archive, 2024), and internal collections, ensuring robust performance across diverse nodule types. The detection outputs lesion bounding boxes, which are then fed into a DenseUNet-based segmentation model (Leotta et al., 2019), originally developed for lung cancer nodule segmentation, to extract volumetric lesion masks.

## 2.3. Lesion Tracking

Lesion tracking ensures consistent monitoring of metastatic lesions across consecutive time points. Rigid registration was employed to align CT scans, as standardized acquisition protocols (such as breath-hold scanning) and minimized anatomical variability prevented large deformations between scans, making rigid registration sufficient for this purpose. The registration process employed anatomical landmarks identified using a deep reinforcement learning technique (Marschner et al., 2022), which detected up to 80 landmarks per scan. These landmarks included key structures such as the spine, lung apex, clavicles, kidneys, and liver. The alignment was achieved using a least-squares rigid registration method based on singular value decomposition (SVD) of the cross-covariance matrix between two sets of 3D landmarks (Arun et al., 1987). Lesions across consecutive scans were matched using an Intersection over Union (IoU) threshold of 0.1. To ensure small lesions were not missed due to potential registration inaccuracies, a minimum lesion diameter of 20 mm was applied for matching purposes, temporarily assigning this value to lesions smaller than 20 mm. These thresholds were determined heuristically. Matched lesions were retained for temporal analysis, while unmatched lesions were excluded to filter out potential false positives. This also accounted for lesions appearing or disappearing between scans, preventing consistent tracking over time. More details on lesion tracking are provided in appendix B.1.

## 2.4. Lesion Pair Identification (Temporal Lesion Pair Classification)

A major contribution of this study is the development of a Temporal Lesion Pair Classifier (TLPC) to address the challenge of identifying clinically significant lesions for RECIST assessment. Since radiologists typically annotate only the most relevant lesions—those exhibiting substantial shrinkage or growth over time—the TLPC ensures that the system focuses on these key lesions, distinguishing them from less relevant ones and enabling clinically meaningful longitudinal analysis. The input to the TLPC are 3D lesion pairs extracted from consectuive CT images. During training, lesion pairs identified by the lesion tracking process described in section 2.3 were used. These were labeled as *Significant*, if they were annotated by the radiologist, and *Insignificant*, if not. The *Insignificant* category includes both false positives detected by the system and true lesions that were not annotated by the radiologist, as only the most clinically relevant lesions were selected for assessment. The TLPC incorporates DenseNet-based feature extractors pre-trained on lung cancer data, obtained from the system described in section 2.2, leveraging prior knowledge for lesion classification. For both training and inference, each lesion in the consecutive study pair was processed separately through parallel DenseNet instances, extracting features that were then

concatenated and passed to a binary classifier. For the classification head of the model's architecture, a design replicating the original DenseNet framework was adopted, adjusted to process concatenated feature maps (lesion pair features). The classification architecture included an Adaptive Average Pooling layer followed by a fully connected layer for binary classification. This lightweight design, with only 4K parameters, was selected due to its efficiency and effective performance in lesion pair classification.

### 2.5. Longitudinal Analysis

To enable the automatic estimation of disease progression, this study incorporated temporal analysis based on an adaption of the RECIST guidelines. The original RECIST classification includes the categories PD, PR, SD, and CR. In this study, the CR category was replaced with a "No Lesions" class, as no patients in the dataset exhibited complete response. This also enabled identification of cases with no significant lesions detected.

The longest diameter of the detected lesions was calculated by identifying boundary voxels within the segmentation mask of each axial slice and determining the maximum pairwise distance between all pairs of boundary points. For temporal analysis, the SoD at each time point was calculated by aggregating the diameters of the selected candidates identified in each image. These values were subsequently used to evaluate disease progression across consecutive study pairs and to calculate objective tumor response at the patient level.

### 3. Data

This study utilized a unique dataset provided by Helsinki University Hospital. It includes longitudinal CT scans from 94 breast cancer patients with lung metastases, with an average of 4 scans per patient. Radiologists annotated up to 15 significant lung lesions per patient, focusing on those exhibiting notable growth or shrinkage. The longest axial diameters of these lesions were recorded to enable temporal tracking of disease progression. As shown in Figure 2, the same lesions were annotated by the radiologist at each timepoint.

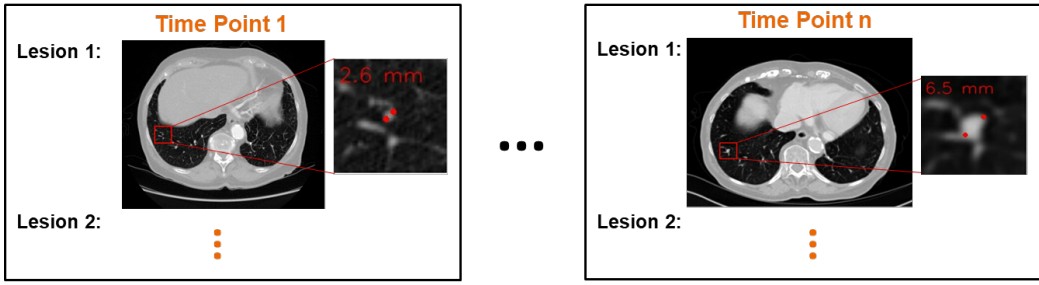

Figure 2: An overview of a patient's lesion data with temporal annotations.

The CT studies were acquired using devices from Siemens Healthineers, GE Medical Systems, and Toshiba. The in-plane spatial resolution ranged from 0.47 mm × 0.47 mm to 0.98 mm × 0.98 mm. Slice thickness varied between 1.5 mm and 5 mm (3.3 ± 0.8 mm). The dataset was randomly divided into training, validation, and test sets at the patient level. Of the 394 studies, 80% were used for training and validation (of which 80% for

training and 20% for validation), and 20% were reserved for testing. This resulted in 60 patients (252 studies) in the training set, 15 patients (63 studies) in the validation set, and 19 patients (79 studies) in the test set. Model and hyperparameter selection was conducted on the validation set.

## 4. Experiments and Results

We performed multiple experiments to assess the performance of individual submodules within the proposed pipeline and to evaluate its overall end-to-end performance for temporal analysis of disease progression. All experiments were conducted on a Tesla V100 GPU (NVIDIA Corporation) with 16 GB of dedicated memory.

**i) Lesion Detection:** The RetinaNet model, originally trained on lung cancer data, was fine-tuned to detect lung lesions from breast cancer metastases. The detection performance was compared to the original pretrained RetinaNet (from the proprietary system mentioned in section 2.2), and to a MONAI implementation of RetinaNet (Cardoso et al., 2022) trained on the LUNA dataset (available as 'Lung Nodule CT Detection' in the MONAI model zoo).

The pretrained and fine-tuned detection models outperformed the MONAI model in terms of sensitivity with the pretrained model achieving the best performance at 0.81, compared to 0.79 for the fine-tuned model and 0.62 for the MONAI model. Due to the complexity of the RetinaNet model and limited data, fine-tuning led to increased detections but also a rise in false positives and false negatives, resulting in fewer true positives. Therefore, the pretrained model, with better overall performance, was selected for longitudinal analysis (further details in appendix A).

**ii) Lesion Tracking:** For lesion tracking, we used the estimated rigid transformation to transform candidate lesions from the first scan into the coordinate system of the second scan. After a hyperparameter search (detailed in the appendix B.2), we selected an IoU threshold of 0.1 and a minimum diameter of 20 mm for lesion matching, where lesions smaller than 20 mm were temporarily assigned this value to prevent small lesions from being missed due to registration inaccuracies, achieving 84% correct matches when applied to the ground truth data on the validation set.

**iii) Lesion Pair Identification:** The TLPC model was trained for 100 epochs using the original DenseNet classifier adapted for binary input with the feature extractor frozen. The preprocessing included clipping image intensities to the range [-1024, 300] HU, linear normalization to [0,1], and resampling to a 0.5 mm isotropic resolution. Data augmentation included random rotations, flipping, zooming, and intensity adjustments to enhance model generalization. Class weights were applied in the weighted cross-entropy loss function to address the imbalance between 829 *significant* and 9145 *insignificant* lesion pairs in the training set. The Adam optimizer (Diederik, 2014) was used for training with a learning rate of $1 \times 10^{-3}$.

The TLPC model achieved an accuracy of 87% and a weighted F1-score of 0.89. The confusion matrix and ROC curve showing the evaluation results of the TLPC model on the validation set in Figure 3 indicate 413 false positives and 64 false negatives, with an AUC of 0.90. These results demonstrate the model's effectiveness in distinguishing significant from insignificant lesion pairs.

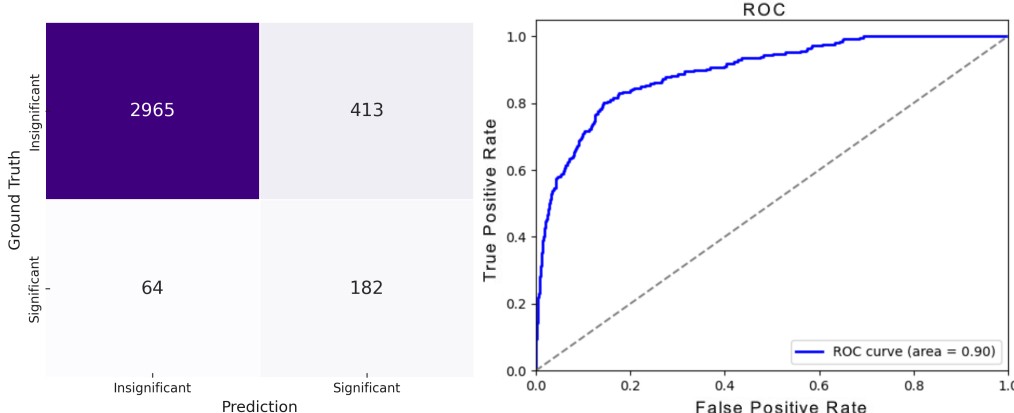

Figure 3: Confusion matrix and ROC curve illustrating the TLPC model's classification performance on the validation set, classifying tracked lesion pairs as significant or insignificant.

**iv) Longitudinal Analysis:** We evaluated the whole pipeline (consisting of lesion detection, segmentation, tracking and identification) for temporal analysis on the test set. We considered both the estimation of disease progression for two consecutive time points and estimation of the objective tumor response on patient-level. We compared our pipeline with a baseline only including lesion detection and segmentation at each time point and calculating the SoD of the detected lesions at each time point.

For the consecutive study pairs, to quantify tumor load dynamics and how well they are aligned with the GT, the relative change in SoD between timepoint 1 and 2 was computed as

$$\Delta_{\text{SoD}} = \frac{\text{SoD}_2 - \text{SoD}_1}{\text{SoD}_1}, \tag{1}$$

where $\text{SoD}_1$ and $\text{SoD}_2$ are the SoDs of timepoint 1 and 2. To assess the similarity between the predicted and ground truth (GT) trends, the absolute difference between their relative changes was calculated. This metric will be referred to as the Relative Change Discrepancy in Sum of Diameters:

$$\text{RCD-SoD} = \left| \Delta_{\text{SoD}_{\text{GT}}} - \Delta_{\text{SoD}_{\text{Prediction}}} \right|. \tag{2}$$

Patient-level tumor response was assessed across multiple time points by combining consecutive time-point analyses. The SoD was calculated at each time point based on lesions detected, matched, and classified as *significant*. When inconsistencies in SoD arose between overlapping study pairs due to differences in lesion selection or classification, the average SoD was used. Finally, predictions and ground truth were evaluated using an adaptation of the RECIST criteria as a multi-class classification problem, categorizing responses into PD, SD, PR, and No Lesions.

*a) Disease progression for consecutive timepoints via SoD:* Figure 4, left, compares the proposed pipeline to the baseline in terms of the RCD-SoD. The proposed approach

achieved a lower RCD-SoD of 10.73 compared to the baseline result of 18.71, representing a 42% reduction (p=0.001 using a Wilcoxon signed-rank test).

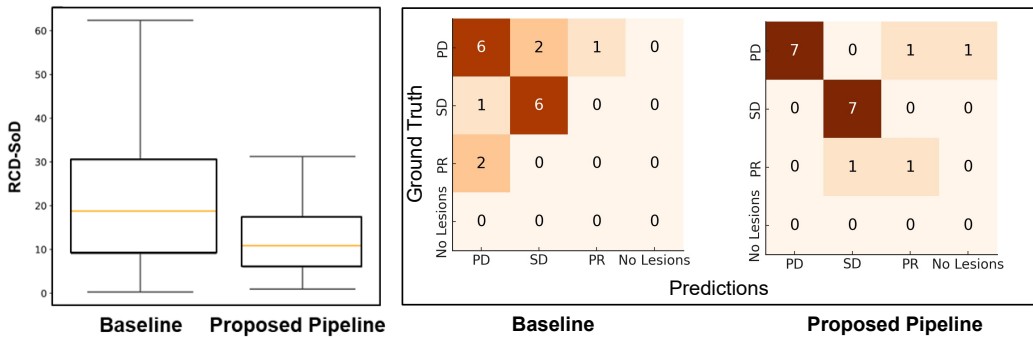

Figure 4: Comparison of RCD-SoD values for the baseline and proposed pipeline for consecutive study pairs (left), Confusion matrices for both pipelines in patient-level evaluation of tumor response classification based on RECIST classes (right).

*b) Disease progression at patient level via RECIST adaptation:* The Baseline and Pair Classification methods were assessed at the patient level, focusing on the full available patient history rather than only two consecutive scans as in previous experiments. This was done by aggregating the SoD values from the pairwise analysis for each time point. Figure 4, right, presents the confusion matrices, while Table 1 summarizes the classification metrics. The proposed pipeline improved all metrics with respect to the baseline results. Most notably, it achieved a higher accuracy of 83% compared with the baseline method (67%).

Table 1: Classification performance of the Baseline and Proposed Pipelines for adapted RECIST scores. The RECIST classes include PR, SD, PD, and No Lesions.

| Method | Accuracy | Precision (Wtd. Avg.) | Recall (Wtd. Avg.) | F1-Score (Wtd. Avg.) |
|---|---|---|---|---|
| Baseline | 0.67 | 0.62 | 0.67 | 0.64 |
| Proposed Pipeline | 0.83 | 0.90 | 0.83 | 0.86 |

## 5. Discussion and Perspectives

This work presents a deep learning-based pipeline for the longitudinal analysis of breast cancer lung metastases. By integrating lesion detection, segmentation, tracking, and identification techniques, the system estimates disease progression and objective tumor response in accordance with an adaptation of the RECIST guidelines. Evaluated on a unique dataset of 94 patients, the pipeline demonstrated significant improvements in tracking accuracy and reduction of false positives.

The evaluation of the TLPC model on the validation set showed a reduction in the number of false positives from 6337 detected lesions by the baseline detection method to 413 false positive lesion pairs. The TLPC model reduced the median RCD-SoD by 42% for consecutive study pairs compared to the baseline and improved tumor response classification accuracy from 67% to 83% at the patient level. These results underscore the system's capability to enhance lesion tracking and provide clinically relevant insights, such as RECIST-based response evaluation.

Key limitations include the small dataset size, which constrained the fine-tuning of the detection model, and the reliance on rigid registration, which is less effective for long-term lesion tracking. Additionally, patient-level response estimation relied on aggregated study-pair results, limiting its precision. Future work should address these by improving lesion tracking across multiple time points, incorporating deformable registration, and handling cases with the emergence of new lesions or the complete disappearance of others.

Overall, this study introduces a fully automated end-to-end pipeline for longitudinal tumor load assessment in breast cancer lung metastases. A key contribution is the TLPC, which distinguishes clinically significant and insignificant lesions. By mimicking RECIST-based decision-making, TLPC enables a clinically meaningful, automated tumor response assessment. This framework offers a structured and clinically relevant solution for longitudinal tumor analysis, addressing a gap not extensively covered by existing approaches.

In conclusion, this study highlights deep learning's potential to automate disease progression estimation and RECIST score calculation, improving tracking consistency and reducing false positives to enhance clinical workflows.

## Acknowledgments

This project has received funding from the European Union's Horizon Europe Research and Innovation Programme under grant agreement No. 101095245.

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

## Appendix A. Lesion Detection

### A.1. Experiment and Results

For fine-tuning of the RetinaNet, CT scans in the training set were resampled to 1 mm isotropic resolution and data augmentation included cropping, flipping, zooming, rotations, and intensity adjustments to enhance generalization. Fine-tuning was performed using SGD with momentum (Sutskever et al., 2013), focal loss, and a maximum of 1000 epochs, selecting the model with the lowest validation loss for evaluation. The optimizer was configured with a learning rate of $1 \times 10^{-2}$, momentum of 0.9, weight decay of $3 \times 10^{-5}$, and Nesterov acceleration.

We compared the lesion detection results of the fine-tuned model with the original model trained on lung cancer data as described in section 2.2. Additionally, we compare to a MONAI's RetinaNet implementation (Cardoso et al., 2022) with publicly available weights trained on the LUNA dataset for lung nodule detection.

The performance of the detection methods was evaluated by measuring sensitivity, along with the counts of true positives (TPs), false positives (FPs), and false negatives (FNs) per scan. To determine TPs, the center coordinates and radii of annotated lesions, calculated as half of their longest axial diameters, were used. Detected candidates' center coordinates were compared to these annotations by calculating the Euclidean distance between their centers. A candidate was classified as a TP if it lay within the spherical region defined by the radius of the annotated lesion. Candidates that did not match any annotated lesions were classified as FPs. Sensitivity was calculated as the proportion of annotated lesions correctly detected, providing a comprehensive assessment of the detection method's performance.

The results of the performance of the lesion detection models are presented in Table 2. Both the pretrained and fine-tuned detection models outperform the MONAI model in terms of maximum sensitivity, defined as the sensitivity achieved when considering all detected lesions without confidence threshold filtering, as well as the number of true positives (TPs). However, the MONAI model exhibits fewer false positives (FPs). The pretrained

RetinaNet model outperformed the fine-tuned version in terms of sensitivity and had less FPs. While the fine-tuned model identified more candidates, it resulted in more FPs without significant improvement in sensitivity. Overall, the pretrained model demonstrated more reliable detection performance on the dataset, likely due to the limited size of the dataset available for fine-tuning. However, it produced a high number of false positives, which may include both insignificant lesions that were not annotated by the radiologist and non-lesions mistakenly detected by the model.

| Metric | MaxS | TP | FP | FN |
|---|---|---|---|---|
| MONAI Model | 0.62 | 357 | 1527 | 214 |
| Pretrained Model | 0.81 | 464 | 6337 | 107 |
| Fine-tuned Model | 0.79 | 449 | 7645 | 122 |

Table 2: Performance comparison for all models measured by the maximum sensitivity (MaxS), true positives (TP), false positives (FP) and false negatives (FN).

## Appendix B. Lesion Tracking

### B.1. Methodology

The alignment between two consecutive CT sancs was achieved using a least-squares rigid registration method based on singular value decomposition (SVD) of the cross-covariance matrix between two sets of 3D landmarks (Arun et al., 1987). Given two sets of $n$ corresponding landmarks, $\mathbf{P}, \mathbf{Q} \in \mathbb{R}^{3 \times n}$ with landmarks $\mathbf{p}_i, \mathbf{q}_i \in \mathbb{R}^3, i = 1, \ldots, n$, the objective is to find a rigid transformation consisting of a rotation matrix $\mathbf{R} \in \mathbb{R}^{3 \times 3}$ and a translation vector $\mathbf{t} \in \mathbb{R}^3$ that minimizes the sum of squared distances between corresponding points:

$$\min_{\mathbf{R}, \mathbf{t}} \sum_{i=1}^{n} \|\mathbf{q}_i - (\mathbf{R}\mathbf{p}_i + \mathbf{t})\|^2. \tag{3}$$

The landmarks are first centered by subtracting their respective centroids $\mathbf{c}_P, \mathbf{c}_Q$. The cross-covariance matrix $\mathbf{K}$ is then computed as

$$\mathbf{K} = \sum_{i=1}^{n} \mathbf{p}_i' \mathbf{q}_i'^{T}, \tag{4}$$

where $\mathbf{p}_i'$ and $\mathbf{q}_i'$ represent the centered landmarks. Applying SVD to $\mathbf{K}$: $\mathbf{K} = \mathbf{U}\mathbf{\Sigma}\mathbf{V}^T$, the optimal rotation matrix is calculated as $\mathbf{R} = \mathbf{V}\mathbf{U}^T$. The translation vector is obtained as $\mathbf{t} = \mathbf{c}_Q - \mathbf{R}\mathbf{c}_P$. The resulting rigid transformation $\mathbf{T}$, comprising $\mathbf{R}$ and $\mathbf{t}$, ensures alignment of corresponding landmarks across consecutive images.

### B.2. Experiment and Results

To evaluate the landmark-based rigid registration, the method was assessed by calculating the Target Registration Error (TRE) using available anatomical landmarks (mentioned in

section 2.3), including the right and left primary bronchi, right and left lung tops, and carina bifurcation. The mean TRE of this approach was compared to a naïve translation-based registration using image centroids. The results showed a 98.2% improvement, with the mean TRE reduced from 430 mm to 7.2 mm.

To determine the most effective criteria for lesion matching, various IoU thresholds and minimum diameters were tested. For each combination, the matched lesions were compared against the ground truth annotations to verify if they corresponded to the same lesion as identified by the radiologist. The criteria yielding the highest number of correct matches in the validation set were selected. After evaluating various criteria, an IoU threshold of 0.1 and a minimum diameter of 20 mm were selected as the optimal parameters, achieving 84% correct matches when applied to the ground truth data on the validation set. These settings consistently identified corresponding annotated lesions across consecutive time points. While an alternative criterion with a minimum diameter of 30 mm (using the same IoU threshold) achieved a slightly higher correct match rate of 85%, it was ultimately not adopted due to also a higher incidence of incorrect matches (33 wrong matches) compared to the 20 mm threshold (4 wrong matches). Table 3 presents results from the evaluation of candidate matching criteria for the various configurations.

Table 3: Evaluation Results for various candidate matching criteria.

| Min. Diameter [mm] | IoU | Percentage of Correct Matches |
|---|---|:---:|
| 10 | 0.1 | 60% |
| 10 | 0.3 | 29% |
| 10 | 0.5 | 13% |
| **20** | **0.1** | **84%** |
| 20 | 0.3 | 79% |
| 20 | 0.5 | 49% |
| **30** | **0.1** | **85%** |
| 30 | 0.3 | 79% |
| 30 | 0.5 | 49% |

Finally we compared our approach against an advanced lesion tracking method proposed by (Vizitiu et al., 2023), which utilizes a multi-scale self-supervised learning framework for lesion tracking. Using the original criteria from that study, the method achieved 38% correct matches. After adapting it to the minimum lesion matching diameter criteria, where all lesions below 20 mm were adjusted accordingly, performance improved to 70% correct matches. However, it remained lower than the accuracy achieved with the proposed approach.

