# OpenReview forum: "Intelligent Lesion Selection: A Novel Method for Longitudinal Assessment of Breast Cancer Lung Metastases"
_MIDL.io/2025/Conference — MIDL 2025 Poster_

### Official Review · Reviewer_DbnN · 2025-02-18

**Confidence:** 4
**Preliminary Rating:** 3
**Final Rating:** 4

**Summary:**

This paper introduces an automated pipeline for the longitudinal assessment of breast cancer lung metastases, integrating lesion detection, segmentation, tracking, and identification. The system leverages a novel Temporal Lesion Pair Classifier (TLPC) to focus on significant lesions, improving the accuracy of disease progression estimation and RECIST-based tumor response classification. Tested on a dataset of 19 patients, the pipeline demonstrated improvements in tracking accuracy compared to baseline methods.

**Strengths:**

- Presents a comprehensive integration of lesion detection, segmentation, tracking, and identification.
- The proposed approach tackles a clinically relevant task.
- The method improves tumor response classification accuracy according to the adapted RECIST classification.
- The paper is well written, well organized, and easy to read.

**Weaknesses:**

- Small Test Dataset Size: the limited test dataset size (19 patients) might affect the generalizability of the results.
- Exclusion of Certain Lesions: lesions smaller than 20mm, as well as appearing/disappearing lesions, were discarded. However, these could have significant clinical value. For example, an appearing lesion could indicate disease progression, changing the response classification from SD to PD. The same applies to lesions between 10mm and 20mm.
- False Positives: Despite improvements, the system still produces a notable number of false positives, which could impact clinical decision-making.

**Detailed Comments:**

- The claim: "These results demonstrate the model’s effectiveness in distinguishing significant from insignificant lesion pairs." in Section 4, iv) is not validated by the results. The model predicts 182 (TP) vs. 413 (FP) for the significant class, meaning that when the model detects a pair as significant, there is a 69% probability that the pair is actually insignificant.
- The metric RCD-SoD, which compares relative changes in SoD between predictions and ground truth, may be misleading. The delta could appear similar even if the SoD was not computed on the same number of lesion pairs. This could have a significant clinical impact.

**Justification Of The Final Rating:**

I would like to thank the authors for their response during the rebuttal period. While they have addressed several points raised in my questions, further analysis of appearing/new lesions remains important for RECIST guidelines. Even though these lesions may be less prevalent, they can significantly impact the RECIST response and should not be overlooked. That said, I recognize that the paper introduces several contributions, making it eligible for acceptance.

**Justification Of The Preliminary Rating:**

The paper presents a well-structured approach for the automated assessment of breast cancer lung metastases, integrating key components for improved tracking and classification. The method shows clear potential, but the small dataset size, exclusion of certain lesion types, and reliance on extensive annotations limit its generalizability and scalability. Addressing these aspects could enhance its clinical impact.

**Questions To Address In The Rebuttal:**

- The reduction in false positives might be largely due to the filtering applied (on non-matched lesions and those smaller than 20mm). However, such lesions are still relevant to RECIST classification, including newly appearing lesions. Would it be possible to provide further evaluation on the model’s capacity to detect these lesion types? A more comprehensive evaluation would strengthen the framework.
- Can you provide more details on the lesion detection module? details on the dataset (multiple institutions?), dataset size, architecture, losses, etc.
- The proposed method relies heavily on radiologists' input, particularly for lesion annotations and defining significant/insignificant pairs. This makes it costly in terms of annotations. Is there a way to reduce dependency on detailed annotations, such as using synthetic approaches to lower annotation requirements?
- For the RCD-SoD metric, is there a way to adapt it to take into consideration the nb of pairs on which the SoD has been computed?

---

> ### Author Response · Authors · 2025-03-07
> **Response to Reviewer 4 (DbnN)**
>
> We sincerely appreciate the reviewer’s feedback and thoughtful evaluation of our work. We are pleased that the integration of multiple components into a structured system was recognized as a strength. Additionally, we value the acknowledgment of our method’s clinical relevance and its contribution to tumor response classification improvements. We address the reviewer’s concerns in detail below.
> 1. We would like to clarify the filtering of non-matched lesions in our pipeline. The baseline method for longitudinal analysis resulted in a high number of false positives (6,337 FP), which included both insignificant lesions not annotated by the radiologist and non-lesions mistakenly detected by the model (added to revised manuscript, Appendix A). Lesion tracking was introduced to enable longitudinal lesion matching and also to reduce false positives, particularly those representing insignificant but real lesions that were not annotated.
> The matched detected lesions in the training set were then used to train the TLPC model, where radiologist-annotated lesions were labeled as "significant", and the rest as "insignificant," mimicking RECIST-like decision-making. This led to a substantial reduction in false positives, decreasing to 413 lesion pairs. While this number remains high for direct clinical application, the results demonstrate the potential of an automated longitudinal analysis system, which we aim to refine further. We also acknowledged that emerging and disappearing lesions were not explicitly analyzed, which is a limitation of this study (highlighted in section 5 in the submitted manuscript). However, their low prevalence in our dataset suggests their exclusion is unlikely to significantly impact the results. Given their potential importance in external validation datasets, we plan to incorporate their analysis in future work to ensure a more comprehensive assessment.
> 2. Regarding the comment on filtering lesions smaller than 20 mm, we acknowledge that our initial wording in the submitted paper may have caused a misunderstanding. After a hyperparameter search (detailed in Appendix B.2), we selected an IoU threshold of 0.1 and a minimum diameter of 20 mm for lesion matching, where lesions smaller than 20 mm were temporarily assigned this value to prevent them from being missed due to registration inaccuracies. We have now clarified this in both the Sections 2.3 and 4.ii.
> 3. We would like to provide more details on the proprietary detection system which was originally designed for lung cancer detection. This module was trained on a dataset of over 6,000 3D chest CT scans from multiple sources. The NLST dataset served as the primary data source, supplemented by the LUNA dataset and an internal collection of cases with proven malignancy status. The system processes 3D CT scans using a RetinaNet for lesion detection. To reduce false positives, detected candidates undergo further classification by an ensemble of a DenseNet and an EfficientNet, which distinguish lesions from non-lesions. Regarding network architectures, we previously cited RetinaNet and have now added citations for EfficientNet and DenseNet in the revised manuscript. The training utilized focal loss for RetinaNet and EfficientNet, while DenseNet3D was trained with weighted cross-entropy. The details about the fine-tuning of the system on our dataset can be found in the Appendix A.
> 4. We acknowledge that our method relies on radiologists' input for training and development. However, we believe that working with expert-annotated data is essential for building clinically meaningful solutions. The annotations and data used in this study are novel and highly valuable as they provide a rare longitudinal perspective on metastatic breast cancer lesions, which is not widely available. Our approach was designed to mimic radiologists' decision-making to identify significant lesions for tumor response assessment. Given the limited availability of such data, we are also exploring the use of synthetic methods to expand our dataset for future development. We appreciate the reviewer’s suggestion and recognize its potential for future enhancements.
> 5. Regarding the comment on integrating the number of lesion pairs into the RCD-SoD metric, we would like to clarify its purpose. The RCD-SoD compares the predicted and ground truth relative changes in SoD, which is a metric designed to reflect overall disease progression over time. Our focus in this study is on treatment response assessment, where we prioritize metrics that capture the overall tumor burden at each time point rather than directly analyzing individual lesions. As a result, even if two cases with different lesion counts have a similar relative change in SoD, this remains valid, as our goal is to capture the broader trend in tumor load dynamics rather than lesion-specific variations. We hope this clarification addresses the concern and have highlighted this in the revised manuscript (Section4, iv).

---

### Official Review · Reviewer_4wpR · 2025-02-19

**Confidence:** 4
**Preliminary Rating:** 2
**Final Rating:** 3

**Summary:**

The paper proposes a framework for the automatic classification of tumor progression/response for breast cancer lung metastases from CT images. The proposed method includes a lesion detection step based on the RetinaNet, a lesion segmentation step based on DenseUNet, lesion tracking between baseline and follow-up scans through rigid registration using also anatomical landmarks, a method to identify significant pairs of lesions using densenet features and the final temporal analysis for the task by incorporating RECIST based criteria.

**Strengths:**

1. The article proposes a complete framework that employs multiple techniques. These are combined in an effective manner leading to an end-to-end longitudinal analysis.
2. The task of tumor progression is a challenging task that can assist the clinical workflow and personalized treatment.

**Weaknesses:**

1. There are concerns regarding the technical/algorithmic novelty of the article. The proposed method combines previously utilized methods for lesion detection (RetinaNet), lesion segmentation (DenseUNet), lesion tracking (rigid registration) and feature extraction (DenseNet) in lesion pairing. These methods are used with only minor changes in a pipeline that ends with the classification task.
2. The evaluation of lesion detection and tracking is limited without presenting a table with additional metrics apart from the match percentage (Appendix). In this regard, the performance of these two crucial steps could be further analyzed.
3. The overall performance in the classification task (Table 1, Figure 4) is not adequately presented. The compared state-of-the-art methods include only the baseline (only including lesion detection and segmentation at each time point and calculating the SoD of the detected lesions at each time point) and not other potential methods. Furthermore, the metrics Precision, Recall and F1-score presented in this table, apart from accuracy (83%), are pretty low ~59% , a value that can prohibit the method’s utilization in the clinical workflow. These values should be further justified or supported by additional experiments.

**Detailed Comments:**

1. In the “2.2. Lesion Detection and Segmentation” section, it is mentioned that internal collections were used for the training. To enhance reproducibility, it is suggested to provide some details such as imaging details (scanner, spacing).
2. For lesion tracking and lesion pair identification, which are important subtasks, a table with metrics will enhance the results’ presentation.
3. On page 12 it is mentioned that “The pretrained RetinaNet model outperformed the fine-tuned version in terms of sensitivity and had less FPs.” Could you please explain in detail what is the finetune version? Is it the RetinaNet without pretraining and with supervised learning?

**Justification Of The Final Rating:**

While the authors expanded the discussion on lesion tracking and registration added value, the novelty of the paper still appears somewhat limited, as it largely integrates pre-existing methods for lesion detection, segmentation, and tracking with small modifications. Nonetheless, the complete framework, including lesion tracking-pairing and “RECIST” incorporation, represents a meaningful technical/clinical contribution. The inclusion of a comparative lesion tracking method and improved registration evaluation is a positive step; however, broader comparisons with state-of-the-art pipelines or additional baselines would strengthen the results. Considering these factors, my recommendation is borderline.

**Justification Of The Preliminary Rating:**

While the article presents a comprehensive pipeline for tumor progression classification by integrating multiple steps, it has two main drawbacks. First, it lacks technical and algorithmic novelty, as its main components are adapted from existing methods with minimal modifications. Second, the evaluation experiments are limited, as the model is compared only against a single baseline method designed by the authors based on previous work. Additionally, despite achieving high accuracy, the method reports notably low F1, Recall, and Precision scores. These low values may result from a calculation error or the small dataset but they could pose a challenge for clinical application.

**Questions To Address In The Rebuttal:**

1. The article lacks technical novelty, as it primarily combines existing methods for lesion detection, segmentation, tracking, and feature extraction with minimal modifications in a pipeline leading to classification. Could you provide additional details on the technical/algorithmic contribution of the study? These additions might enhance the presentation of the methodology.
2. The evaluation of lesion detection and tracking is limited, as it only reports the match percentage (Appendix) without additional metrics. A more detailed analysis of these key steps would strengthen the study.
3. The overall classification performance (Table 1, Figure 4) is not adequately presented. The comparison includes only a baseline method—limited to lesion detection, segmentation, and SoD calculation—without evaluating alternative approaches. Additionally, the reported Precision, Recall, and F1-score (~59%) are significantly lower than accuracy (83%), which may hinder clinical applicability. These low values may result from a calculation error or they would pose a challenge for clinical application. So they should be further justified or supported by additional experiments.

---

> ### Author Response · Authors · 2025-03-07
> **Response to Reviewer 3 (4wpR)**
>
> We sincerely appreciate the reviewer’s feedback and recognition of our study. We are pleased that the integration of multiple techniques into a structured framework was seen as a strength. Additionally, we value the acknowledgment of the challenges in tumor progression assessment and the potential of our approach to contribute to clinical workflows. Below, we address the reviewer’s comments in detail.
> 1. We acknowledge the reviewer’s concern regarding technical novelty and would like to clarify the key contributions of our study. Our work extends beyond combining existing methods by introducing a fully automated end-to-end pipeline that integrates detection, segmentation, tracking, and classification for longitudinal tumor load assessment. Unlike conventional approaches, our pipeline is specifically designed for breast cancer lung metastases, ensuring clinically relevant lesion selection and improving tumor response classification.
> A major technical contribution is the novel TLPC. Existing lesion detection and tracking models cannot be directly applied for RECIST-like analysis, as they do not differentiate clinically significant and insignificant lesions. TLPC addresses this by learning to identify the significant lesions for tumor burden assessment, mimicking RECIST-based decision-making rather than relying on predefined rule-based methods. This advancement is essential for conducting an automated and clinically meaningful RECIST-like evaluation. By combining these elements, our study provides a structured, and clinically relevant solution for automated longitudinal tumor assessment, an area that has not been extensively addressed by existing approaches. We clarified our contributions in section 5 of the revised manuscript.
> 2. We understand the concern regarding the limitations of rigid registration. We chose this approach because minimal anatomical variability between consecutive scans reduced the need for deformable registration. Our results demonstrated that rigid registration was sufficient for accurate lesion tracking, achieving 84% correct matches against ground truth annotations. The goal was not voxel-level accuracy but robust lesion matching across timepoints.
> To further evaluate rigid registration, we assessed the method by calculating the Target Registration Error (TRE) using available anatomical landmarks (added to the revised manuscript, Appendix B.2). The mean TRE of this approach was compared to a naïve translation-based registration using image centroids. The results showed a 98.2% improvement, with the mean TRE to 7.2 mm. While the mean TRE of 7.2 mm may seem high, our method does not rely on voxel-perfect alignment. Instead, we incorporate a diameter-based search region to account for minor misalignments while ensuring reliable lesion correspondence.
> For comparison with other lesion tracking methods, we evaluated our approach against another Lesion Tracking method (added in the revised manuscript, Appendix B.2). Using the original criteria from that study, we achieved only 38% correct matches. However, after adapting the method to our minimum lesion matching diameter criteria, performance improved to 70% correct matches—still lower than our method. We acknowledge that alternative approaches, such as deformable registration or point tracking methods, may further improve lesion alignment, particularly for long-term follow-ups where anatomical changes are more pronounced. Investigating these techniques remains an important direction for future work.
> 3. We acknowledge the lack of comparison with an alternative pipeline and clarify that our experiments aimed to demonstrate both the potential of the proposed pipeline and the effectiveness of TLPC. While learning tumor progression directly from images is an alternative, we find it clinically uninterpretable and limited by data constraints. In our study, we compared the TLPC-based pipeline (using only significant lesion pairs) with a baseline using all detected lesions. Additionally, we have now developed a preliminary pipeline excluding TLPC, which uses only the matched lesions for longitudinal analysis. This resulted in a higher median RCD-SoD of 16.82 compared to 10.73 by our method (baseline: 18.71), confirming our full pipeline, including TLPC, achieves superior performance.
> We appreciate the reviewer’s question regarding Table 1. The low averages for precision, recall, and F1-score were due to class imbalance, particularly the absence of the "no lesions" class in the ground truth, which affected overall averages. To address this, we updated the table to report weighted average values, ensuring a more accurate representation of the model’s performance. The revised values better reflect our results.
> 4. The detection model was fine-tuned on the training set used in this study, which consists of 3D CT scans containing lung lesions from breast cancer metastases. Further details on the training configuration are provided in Appendix A.

---

### Official Review · Reviewer_hmhj · 2025-02-25

**Confidence:** 4
**Preliminary Rating:** 3

**Summary:**

The paper presents an automated pipeline for tracking breast cancer lung metastases using CT scans, integrating lesion detection, segmentation, tracking, and a Temporal Lesion Pair Classifier (TLPC). Evaluated on 94 patients, it reduced tumor progression discrepancy by 42% and improved tumor response classification by 22%. The TLPC achieved 87% accuracy and an AUC of 0.90, enhancing RECIST-based assessment. By automating lesion selection and tracking, this method improves clinical decision-making and reduces radiologists' workload.

**Strengths:**

- Combines lesion detection, segmentation, tracking, and classification, improving tumor progression analysis.

- Temporal Lesion Pair Classifier (TLPC): Enhances RECIST-based assessment, achieving 87% accuracy and AUC of 0.90.

- Reduces manual workload and inter-reader variability, making it highly relevant for radiologists.

- Thorough Experimental Design: Evaluates performance across 94 patients, addressing real-world challenges.

- Strong Literature Review: Effectively contrasts contributions against prior work, showing clear advancements.

-  Clear Writing & Structure: Well-organized, follows scientific principles, and presents results logically.

- High Potential for Adoption: While not necessarily state-of-the-art, the approach has strong practical value in medical imaging.

**Weaknesses:**

- Based on **94 patients**, limiting generalizability; larger multi-center data would strengthen findings.
- May not handle **long-term anatomical changes**; deformable registration could improve tracking.
- TLPC is trained on **clinically significant lesions**, possibly overlooking subtle but relevant changes.
-  Tested only on **hospital data**, raising concerns about generalizability across scanners and populations.
-  RECIST-based assessment may introduce **inconsistencies** due to varying lesion selection.
-  No direct **comparison to state-of-the-art lesion tracking models**, limiting contextual performance insights.

**Detailed Comments:**

- Clarify TLPC Training: Specify if all annotated lesions were used or filtered by clinical significance.
- External Validation: Testing on an independent dataset would improve generalizability.
- Compare to State-of-the-Art: Directly benchmark against other lesion tracking models.
- Consider Deformable Registration: May improve long-term lesion tracking over rigid registration.
- Handle Lesion Disappearance: Clarify how the method deals with new or vanishing lesions.
- Improve Figures: Some ROC curves and confusion matrices could be clearer.
- Ensure Terminology Consistency: Standardize terms like TLPC, RECIST adaptation, and lesion tracking.

**Justification Of The Preliminary Rating:**

- Clarify TLPC Training: Specify if all annotated lesions were used or filtered by clinical significance.
- External Validation: Testing on an independent dataset would improve generalizability.
- Compare to State-of-the-Art: Directly benchmark against other lesion tracking models.
- Consider Deformable Registration: May improve long-term lesion tracking over rigid registration.
- Handle Lesion Disappearance: Clarify how the method deals with new or vanishing lesions.
- Improve Figures: Some ROC curves and confusion matrices could be clearer.
- Ensure Terminology Consistency: Standardize terms like TLPC, RECIST adaptation, and lesion tracking.

**Questions To Address In The Rebuttal:**

NA

**Special Issue:**

No

---

> ### Author Response · Authors · 2025-03-07
> **Response to Reviewer 2 (hmhj)**
>
> We sincerely thank the reviewer for their valuable feedback and appreciation of our work. We are pleased that the integration of lesion detection, segmentation, tracking, and classification was recognized as a strength in improving tumor progression analysis. We also appreciate the acknowledgment of our Temporal Lesion Pair Classifier (TLPC) and its contribution to RECIST-based assessment. Furthermore, we are grateful for the positive remarks on our experimental design, literature review, and manuscript clarity, as well as the recognition of the practical value of our approach in clinical settings. We address the reviewer’s individual concerns below.
> 1. We would like to clarify the annotation process used in our study. Due to the time-consuming nature of lesion annotation, as stated in the submitted paper, the radiologist focused only on clinically significant lesions—those showing considerable shrinkage or growth. As a result, not all lesions were annotated. This selective annotation approach aligns with real-world clinical practice, where radiologists prioritize lesions that influence disease progression assessment.
> Regarding the training of the TLPC model, we first applied the lesion detection model to the training set, followed by lesion tracking to obtain lesion pairs. Lesion pairs that were annotated by the radiologist were labeled as "significant," while lesion pairs without annotations were marked as "insignificant." This approach allowed the model to learn to distinguish lesions that are most relevant for tumor progression assessment, aligning with how radiologists prioritize such lesions in clinical practice.
> 3. We acknowledge that the dataset size poses challenges for generalizability and understand the concern regarding validation on an internal dataset. To our knowledge, no publicly available dataset fully aligns with our study’s scope—longitudinal CT scans of metastatic breast cancer to the lung. We fully agree that external, possibly multi-center, validation is essential for assessing generalizability. This remains a key objective of the project, and we will conduct such validation as soon as data (so far planned for two additional centers) becomes available. Ensuring broader validation across different scanners and clinical settings is an important future direction.
> 4. We understand the concern regarding the limitations of rigid registration. We chose this approach because standardized acquisition protocols and minimal anatomical variability between consecutive scans reduced the need for deformable registration. Our results demonstrated that rigid registration was sufficient for accurate lesion tracking, achieving 84% correct matches against ground truth annotations. The goal was not voxel-level accuracy but robust lesion matching across timepoints.
> To further evaluate rigid registration, we assessed the method by calculating the Target Registration Error (TRE) using available anatomical landmarks (added to the revised manuscript, Appendix B.2). The mean TRE of this approach was compared to a naïve translation-based registration using image centroids. The results showed a 98.2% improvement, with the mean TRE reduced to 7.2 mm. While the mean TRE of 7.2 mm may seem high, our method does not rely on voxel-perfect alignment. Instead, we incorporate a diameter-based search region to account for minor misalignments while ensuring reliable lesion correspondence.
> For comparison with other lesion tracking methods, we evaluated our approach against another Lesion Tracking method (cited in Appendix B.2 of the revised manuscript ). Using the original criteria from that study, we achieved only 38% correct matches. However, after adapting the method to our minimum lesion matching diameter criteria, performance improved to 70% correct matches—still lower than the 85% achieved with our landmark-based registration (added to the revised manuscript, Appendix B.2).
> We acknowledge that alternative approaches, such as deformable registration or point tracking methods, may further improve lesion alignment, particularly for long-term follow-ups where anatomical changes are more pronounced. Investigating these techniques remains an important direction for future work.
> 5. Emerging and disappearing lesions were not explicitly analyzed in this study, which we acknowledge as a limitation  (mentioned in section 5 of the submitted manuscript).   However, given their low prevalence in our dataset, their exclusion is unlikely to significantly impact our results. We recognize their clinical  importance, particularly in external validation data, where their occurrence may be higher. To address this, we plan to incorporate the analysis of these lesion types in future work, ensuring a more comprehensive assessment of tumor response dynamics.
> 5. We appreciate the feedback regarding the figures (figures 3 and 4) and terminology and have addressed these issues in the revised manuscript.

---

### Official Review · Reviewer_Xa4T · 2025-02-25

**Confidence:** 3
**Preliminary Rating:** 4
**Recommendation:** Oral
**Final Rating:** 4

**Summary:**

In this study, the authors propose a novel method for the longitudinal analysis of breast cancer lung metastasis for automated tumor response assessment. Their proposed method shows promising results with less discrepancy in median tumor size progression and higher accuracy in tumor response classification.

**Strengths:**

- The paper tackles a strong clinical challenge of automating the longitudinal assessment of tumor response using established clinical standards like RECIST.
- The proposed method dramatically outperforms the baseline model.
- The experiments are well-designed and each component is validated thoroughly.

**Weaknesses:**

- The proposed method is only validated on an internal test set. Further validation using external data to needed to evaluate the method's generalizability.
- The CT scans were registered using rigid transformation rather than deformable registration. While this may be sufficient as per the acquisition protocol, small changes in anatomy could impact lesion tracking.
- The study lacks statistical comparisons with the baseline method.

**Detailed Comments:**

Please see weaknesses above.

**Justification Of The Final Rating:**

The authors have addressed all of my questions, and the addition of statistical analysis further strengthens the findings. While I agree with the concerns raised by other reviewers (small dataset size, limited generalizability, etc.), I believe the contributions of this work are sufficient for acceptance as a full paper.

**Justification Of The Preliminary Rating:**

The paper proposes a novel method for automating the longitudinal analysis of breast cancer lung metastases, outperforming the baseline method. Despite the limited validation only on internal data, the work has a strong clinical motivation and the results show promising implications in clinical settings.

**Questions To Address In The Rebuttal:**

Please see weaknesses above.

---

> ### Author Response · Authors · 2025-03-07
> **Response to Reviewer 1 (Xa4T)**
>
> We sincerely thank the reviewer for their thoughtful feedback and for recognizing the clinical significance of our work. We appreciate the acknowledgment that the task of our study is challenging, and we are grateful for the recognition that our experimental design is well-structured and thoroughly validated, outperforming the baseline. We have carefully considered the reviewer’s comments and provide detailed responses below.
> 1. We acknowledge the concern regarding validation of only an internal dataset. To our knowledge, no publicly available dataset fully aligns with our study’s scope—longitudinal CT scans of metastatic breast cancer to the lung. While the NLST dataset, which includes multiple CT scans per patient, might have been a potential option for external validation, it was previously used for training the proprietary lesion detection model in our pipeline and, therefore, cannot be used for validation. We fully agree that external, possibly multi-center, validation is essential for assessing generalizability. This remains a key objective of the project, and we will conduct such validation as soon as data (so far planned for two additional centers) becomes available. Ensuring broader validation across different scanners and clinical settings is an important future direction.
> 2. We understand the concern regarding the limitations of rigid registration. We chose this approach because standardized acquisition protocols and minimal anatomical variability between consecutive scans reduced the need for deformable registration. Our results demonstrated that rigid registration was sufficient for accurate lesion tracking, achieving 84% correct matches against ground truth annotations. The goal was not voxel-level accuracy but robust lesion matching across timepoints.
> To further evaluate rigid registration, we assessed the method by calculating the Target Registration Error (TRE) using available anatomical landmarks (added to the revised manuscript, Appendix B.2), including the right and left primary bronchi, right and left lung tops, and carina bifurcation. The mean TRE of this approach was compared to a naïve translation-based registration using image centroids. The results showed a 98.2% improvement, with the mean TRE reduced from 430 mm to 7.2 mm. While the final mean TRE of 7.2 mm may seem high, our method does not rely on voxel-perfect alignment. Instead, we incorporate a diameter-based search region to account for minor registration discrepancies while ensuring reliable lesion correspondence.
> For comparison with other lesion tracking methods, we evaluated our approach against a Multi-Scale Self-Supervised Learning method for Longitudinal Lesion Tracking (cited in Appendix B.2 of the revised manuscript). Using the original criteria from that study, we achieved only 38% correct matches. However, after adapting the method to our minimum lesion matching diameter criteria (setting lesions smaller than 20 mm to 20 mm), performance improved to 70% correct matches—still lower than the 85% achieved with our landmark-based registration (added to the revised manuscript, Appendix B.2).
> We acknowledge that alternative approaches, such as deformable registration or point tracking methods, may further improve lesion alignment, particularly for long-term follow-ups where anatomical changes are more pronounced. Investigating these techniques remains an important direction for future work.
> 3. We agree that providing statistical comparisons with the baseline approach is important and have added this analysis in section 4 of the paper. To achieve this, we compared the Relative Change Discrepancy in Sum of Diameters (RCD-SoD) values between the baseline and our proposed approach. Since the Shapiro-Wilk test indicated that the data is not normally distributed (p = 1.62e-8), we used the Wilcoxon signed-rank test to assess statistical significance. The test resulted in a p-value of 0.001, confirming that the observed improvements of our method over the baseline are statistically significant. We have now highlighted this result in section 4, iv in the revised manuscript.

---

### Author Rebuttal · Authors · 2025-03-07

**Rebuttal:**

We sincerely thank the reviewers for their constructive feedback and valuable insights. We appreciate their recognition of our experimental design, validation of each pipeline component, and the clinical relevance of our approach. Below, we summarize the most critical concerns raised, while detailed responses are provided in the individual reviewer comments. Specific changes in the manuscript are highlighted in purple color for clarity.
One key concern raised by multiple reviewers was the lack of external validation. To our knowledge, no publicly available dataset provides longitudinal CT scans of metastatic breast cancer to the lung. However, multi-center validation is a planned future step in this project.
Regarding lesion tracking, our evaluation confirmed that rigid registration provided sufficient lesion tracking, achieving 84% correct matches against ground truth. We used a diameter-based search region to handle minor misalignments.  Rigid registration was chosen for its efficiency and effectiveness, and future work will explore deformable and point-tracking methods. To further assess our approach, we have now evaluated registration and compared our lesion tracking method to an alternative approach in the revised manuscript (Appendix B.2).
The analysis of emerging and disappearing lesions was not explicitly included in this study, which we acknowledged as a limitation. However, given their low prevalence in our dataset, their exclusion is unlikely to significantly impact the results. We plan to incorporate their analysis in future work.
Regarding the novelty of our approach, most existing models cannot directly perform RECIST-like analysis as they do not distinguish between significant and insignificant lesions. Our Temporal Lesion Pair Classifier (TLPC) addresses this limitation by learning to differentiate lesion pairs based on clinical relevance. Since radiologists annotated only significant lesions, the model was trained to prioritize these, enabling an automated and clinically meaningful tumor response assessment.
Our contribution lies in the fully automated integration of detection, segmentation, tracking, and classification, as well as the novel TLPC model, addressing a clinically relevant challenge and improving tumor response classification.
We appreciate the reviewers' thoughtful feedback and believe these clarifications strengthen our study. Thank you for your consideration.

**Supporting Material:**

/attachment/3130ac654bea84fb624a86002572f9706ea3af70.pdf

---

### Meta-Review · Area_Chair_5qKV · 2025-03-23

**Recommendation:** Accept (Poster)
**Confidence:** 5

**Metareview:**

Summary:
The authors propose a framework for the longitudinal assessment of lung metastases in breast cancer patients. They combine lesion detection, segmentation, and tracking to identify longitudinal changes. A  Temporal Lesion Pair Classifier (TLPC) identified significant lesions. Reviewer comments were sufficiently addressed. However, there are some concerns regarding the small dataset size and the validity of the statistical results as a consequence.

Metareviewer decision:
Accept